# Microclimate and Genotype Impact on Nutritional and Antinutritional Quality of Locally Adapted Landraces of Common Bean (*Phaseolus vulgaris* L.)

**DOI:** 10.3390/foods12061119

**Published:** 2023-03-07

**Authors:** Irene Bosmali, Ilias Giannenas, Styliani Christophoridou, Christos G. Ganos, Aggelos Papadopoulos, Fokion Papathanasiou, Alexandros Kolonas, Olga Gortzi

**Affiliations:** 1Institute of Applied Biosciences, CERTH, 6th km Charilaou-Thermis Road, Thermi, 57001 Thessaloniki, Greece; 2Laboratory of Nutrition, Faculty of Veterinary Medicine, Aristotle University of Thessaloniki, 54124 Thessaloniki, Greece; 3Laboratory of Food Safety and Quality Assurance, Department of Agriculture, Crop Production and Rural Environment, University of Thessaly, 38446 Volos, Greece; 4Department of Agriculture, School of Agricultural Sciences, University of Western Macedonia, 53100 Florina, Greece

**Keywords:** common bean, *Phaseolus vulgaris* L., nutritional, antinutritional factors

## Abstract

This study aimed to assess the impact of genotype, location, and type of cultivation (organic) on the nutrient and anti-nutrient components of seven large-seeded bean (*Phaseolus vulgaris* L.) populations. All genotypes were cultivated during 2014 and 2015 in randomized complete block (RCB) experimental designs in three areas of the Prespa region (Pili, Patoulidio, Agios Germanos) in Greece. Particularly, total protein (18.79–23.93%), fiber (7.77–12%), starch (40.14–55.26%), and fat (1.84–2.58%) contents were analyzed and showed significant differences. In order to assess mineral content, firstly, the total ash percentage (4.31% to 5.20%) and secondly, trace elements and heavy metals were determined. The concentrations of identified inorganic metals showed large variations. The total phenolic content of the samples varied from 0.18 to 0.29 mg/g gallic acid equivalent (GAE). A major limitation of increasing the use of grain legumes as feed is the presence of diverse compounds in their grain, commonly referred to as antinutritional factors, and these are mainly trypsin inhibitors. Trypsin inhibitor levels were evaluated, with results varying from 21.8 to 138.5 TIU/g. Pili 2014 and 2015 were differently associated regarding the year of cultivation. Pili 2015 location was also very closely associated with the Patoulidio region, whereas Agios Germanos and Pili 2014 were the most different in terms of nutritional and antinutritional content.

## 1. Introduction

The common bean (*Phaseolus vulgaris* L.) is a plant of the legume family (Fabaceae), and it represents the most widely distributed and consumed legume species [1]. Bean production accounts for 46% of the overall production of legumes, almost twice that of chickpea (22%), which is the second-most produced grain legume. This is because the common bean is an essential source of dietary proteins and carbohydrates for many people worldwide, along with beneficial effects on human health in terms of anti-diabetic, cardioprotective, and protective effects against various types of cancer [2]. Common beans are highly nutritious, having almost twice the protein content of cereals [3], lower fat than other legumes, and high amounts of minerals [4]. *Phaseolus vulgaris* L. represents an important source of bioactive compounds in the Mediterranean diet, including phenolic compounds that exhibit antimutagenic, anticarcinogenic, antioxidant, and antiapoptotic properties, improvement of endothelial functions, and inhibition of cell proliferation activities. In addition, *Phaseolus vulgaris* L. contains an alpha-amylase inhibitor, which has been tested in clinical studies for its ability to promote weight loss and glycemic control. In many developing countries, especially in those of Latin America, Eastern, and Southern Africa, the common bean is a principal source of proteins in the diet of rural and urban low-income populations [5]. Since their introduction in Greece (late 16th century CE), *Phaseolus* beans have become well adapted, creating a long tradition in cultivation and consumption, which resulted in a generation of locally domesticated populations depending on the microclimate of the specific area. The proteins of beans have a high lysine content, while they only have low amounts of sulfur-containing amino acids such as methionine and cysteine [6,7]. However, dry beans possess antinutritional properties such as enzyme (trypsin, chymotrypsin, a-amylase) inhibitors and phytic acid, a compound that reduces the bioavailability of multivalent minerals, saponins, toxic factors, and lectins [7]. Among many antinutritional factors, trypsin inhibitor proteins were found to be one of the most prominent antinutritional constituents. In the dry bean, it acts as a protective agent for the seed. However, trypsin can reduce the total protein intake (digestibility) in humans by inhibiting the activity of proteinase enzymes [8].

The chemical and nutritional compositions of dry beans are influenced by the microclimatic conditions of the cultivation area [9], genotype, storage, and processing conditions [10]. Nutritional factors (protein, starch, amylose, amylopectin, sucrose, citric acid, and malic acid) have been shown to vary significantly depending on the cultivation location, with less significant variation resulting from different varieties. The main variability factors between cultivation locations were found to be protein and starch content, while the least variable was found to be amylopectin content [11]. Barampama and Simard [12] reported significant differences among segregated populations of bean seeds for zinc, phosphorus, iron, and calcium accumulation, while market classes of dry edible beans varied widely for seed coating color, seed size, and texture. Colored bean types have been reported to be higher in antioxidants than white-seeded types [13]. The quantity and activity of trypsin inhibitors are influenced by the specific genotype, cultivation technique, processing, etc. Nevertheless, it is well established that protease inhibitors can have preventive effects against cancer. Lastly, antinutritional factors in beans vary depending on the microclimatic conditions, cultivation location, and variety [8].

*Phaseolus* beans have been subjected to multiple domestications in time and space [14,15]. Today, *Phaseolus* beans are cultivated in almost all continents, in various environments, ranging from tropical, subtropical to temperate climates. Beans quickly adapt to new environments generating extremely diverse landraces in morphological variability, nutritional quality, cultivation requirements, etc. Broad adoption of the species in terms of cultivation requirements, consumer acceptability, and protein nutritional value has made the common bean the most widely grown legume for direct human consumption [5]. Adaptation to new environments resulted in extensive biodiversity compared to the original seed. According to Beebe et al. [16] most bean types cultivated in Europe belong to the Andean genetic pool (large-seeded race Nueva Granada). Long-term cultivation at distinct microenvironments, combined with extensive genetic heterogeneity, led to various landraces with particular genetic and morphological traits [17]. This revival of landraces is important for the added value of local products [18]. Adaptation results in improved varieties with valuable traits in terms of yield and stability [19,20], as well as disease resistance and stress tolerance [21]. Moreover, their product has desirable local market quality traits (i.e., easy cooking, tasteful, thin peel) [22,23].

In the last decades, landraces have been gradually replaced by imported cultivars in response to market requests. However, several local populations still survive on-farm in marginal areas of several European countries, such as Spain, Italy, the Netherlands, and Switzerland [24]. In Greece and neighboring countries, dry bean local populations are still cultivated with traditional methods (e.g., harvesting by hands). Compared to commercial varieties, these populations are less productive and more variable but better adapted to the specific pedoclimatic conditions of these areas [20,25]. Such local populations constitute valuable genetic resources that could be commercially exploited following appropriate evaluation and selection. This is a significant factor for organic agriculture as the objective is to use local varieties or populations characterized by high adaptability to low input system [26]. Organic farmers can profit from the genetic material’s physiological and qualitative characteristics adapted to local conditions with possible tolerance to diseases and weed competition [25]. Consumer preference for high-quality products with suitable physicochemical characteristics is also essential when selecting cultivars adapted to organic farming. Therefore, it is a fundamental concern of an organic management system to evaluate and choose varieties or locally adapted landraces that combine high yields with suitable quality properties under low inputs [27].

The Prespa Lakes are two tectonic lakes, Megali (Big) Prespa, which is located on the tripoint of Greece, North Macedonia, and Albania, and Mikri (Small) Prespa, which belongs mainly to Greece and is characterized by rich local flora. Archeological records indicate that the Prespa basin has been inhabited for more than six thousand years, while currently, on the Greek side of the region, the lakes are encircled by extensive farmland for the cultivation of beans. The Prespa National Park area (PNP), in 1975, was designated a “Landscape of Outstanding Natural Beauty”, and in 2009, several areas of the PNP were designated “Protected Natural Formation or Landscape Elements” [28].

The objective of this research was to investigate the effect of genotype, agronomical conditions (soil type, fertilization, etc.), and environmental (microclimate, specific weather conditions) differences in nutritional and antinutritional properties of locally adapted *Phaseolus vulgaris* landraces in order to evaluate their performance under different agroecological environments in the region of Lake Prespa Greece.

## 2. Materials and Methods

### 2.1. Plant Material

Seven landraces of the common bean (Figure 1a), *Phaseolus vulgaris* L. (Figure 1b), of climbing type IV, collected over five years in traditional areas of common cultivation in Northern Greece and North Macedonia, were used (Table 1). Each landrace, cultivated for over 30 years by each farmer avoiding seed mixing, was originally collected (500 g sample) from farmer’s stocks. The populations A, B, and D were collected around Prespa Lakes (A: Agios Germanos, B: Plati, D: Laimos), population C from neighboring North Macedonia (C: Nakolets) and landraces E, F, and G also from Northern Greece (E: Chrisoupoli, F: Kastoria, G: Florina).

### 2.2. Experimental Design

The seven genotypes were planted at three locations in the traditional area of common bean cultivation close to Lakes Prespa, Greece. i.e., Pili (40°46′ N 21°02′ E latitude, 858 m altitude, 11.8 °C mean temperature, average rainfall 650 mm, cultivation years 2014 and 2015), Patoulidio (40°49′ N 21°06′ E latitude, 859 m altitude, 11.5 °C mean temperature, average rainfall 680 mm, cultivation year 2015) and Agios Germanos (40°50′ N 21°09′ E latitude, 960 m altitude, 11.3 °C mean temperature, average rainfall 700 mm, cultivation year 2015). Soil type, pH, calcium carbonate (CaCO_3_), electrical conductivity (EC), and organic matter of each of the experimental locations are shown in Table 2. All experiments were set up on fully organic-certified fields. Pili was considered a well-irrigated and fully fertilized site, whereas Patoulidio and Agios Germanos were considered intermediate irrigated and fertilized sites due to the scarcity of irrigation water during the maturation period of the bean crop. Irrigation in the Pili site was performed once a week with a drip irrigation system according to the needs of the crop up until physiological maturity of the crop, whereas in Patoulidio and Agios Germanos, irrigation was performed with furrow flooding once a week and just before and during the seed filling stage every two weeks due to scarcity of irrigation water. Fertilization in Pili was performed every year, while in Patoulidio and Agios Germanos, every two years, using 50 tons per hectare of dry cattle manure in all fields. The agronomic practices in all sites followed those of organic agriculture recommended for the crop in this region. The field trials were arranged as randomized complete blocks with four replications. Each replication consisted of seven plots (one for each landrace) and four 3 m rows, with a plant-to-plant distance of 60 cm and a row-to-row distance of 70 cm. Border plants were omitted in harvest, and the final seed product was taken from the middle plants. All experiments in both years were sown in mid-May and harvested from early October until mid-November, depending on the physiological maturity of the crop and the weather conditions.

### 2.3. Pre Treatment

Samples (100 g) were collected from each plot (7 plots) at each location (3 sites) and for each genotype (A, B, C, D, E, F, and G). The samples of the seven different plots were mixed in order to have a representative sample; thus, the resulting number of samples (genotype x site) was 21. Each of the samples was cut in half to increase the surface and then freeze-dried for 24–36 h to measure the moisture content and safely store the dried samples at −40 °C. For further analysis, the freeze-dried samples were crushed and ground in a Polymix dispersing device with a sieve opening of 2 mm firstly, 1.5 mm afterward, and 1 mm finally. The ground material was subjected to nutritional analysis in terms of protein, lipid, fiber, starch, phenolic, and ash content and antinutritional analysis in terms of trypsin inhibitor activity.

### 2.4. Nutritional Evaluation

*Phaseolus* samples were analyzed according to the Weende system for moisture, crude protein (CP), ether extract (EE), crude fiber (CF), and ash by FOSS NIRS NIR DA1650 by FOSS analytical equipment (FOSS, Hilleroed, Denmark) that was respectively calibrated. Near-infrared spectroscopy (NIRS) has taken several steps toward ultimate performance, combining exceptional accuracy across a broad wavelength range of 400–2500 nm with full compatibility with all existing and future FOSS instruments, designed for use in the laboratory or feed mills, ideal for routine control of intake for optimal use of raw materials, routine production control for improved efficiency and economy, final product monitoring on diverse control parameters, as previously described by Giannenas et al. [29]. In order to ensure the results obtained by the NIRS, several of the samples were subjected to chemical analysis using the same methodology for each nutrient or energy. Dry matter analysis was performed by drying in an oven at 135 °C for 2 h (method 930.15; AOAC, 2005 [30]). Analysis of CP was determined by the Kjehdahl method (method 2001.11; AOAC, 2005 [30]). The ether extract was analyzed after extracting crude fat with ether to a Soxtherm 2000 apparatus by Gerhardt (method 920.39; AOAC, 2005). The crude fiber was determined by using a Fibertec system M 1017 hot extractor and 1018 cold extractor by Tecator (Barcelona, Spain). Ash was determined by ignition at 550 °C for 6 h until all carbon has been removed (method 942.05; AOAC, 2005 [30]). Gross energy (GE) was determined with a bomb calorimeter (Julius Peters, Berlin, Germany).

#### 2.4.1. Total Phenolic Content Determination

Folin–Ciocalteu assay was used for the determination of the total phenol content of *Phaseolus vulgaris* seeds extract following the procedure described by Ainsworth and Gillespie [31]. Briefly, 0.1 mL of each of the diluted test samples was added to 0.5 mL of 10% (*v*/*v*) Folin–Ciocalteu reagent, and after 3–8 min, 0.4 mL of 35% (*w*/*w*) sodium carbonate solution was added to the mixture. The absorbance of the reaction mixture was measured at 765 nm after incubation for 30 min at room temperature. A set of standard solutions of gallic acid (0, 1, 2, 3, 4, 5, 6, 7, 8, 9, and 10 mg/L) were prepared in the same manner as described earlier. Incubated for 60 min at room temperature and the absorbance for test and standard solutions was determined against the reagent blank at 765 nm with a Shimadzu UV-2550PC Ultraviolet (UV) /Visible spectrophotometer (Shimadzu, Kyoto, Japan). Total phenolic content was expressed as mg of gallic acid equivalent (GAE)/g of extract.

#### 2.4.2. Ash Content

The atomic absorption method [32] was used to determine the ash content of *Phaseolus vulgaris* seeds after their carbonation had preceded. More specifically, 4 g of each sample was carbonated and cremated in a capsule at 550 °C for 48 h. After that, the capsules were placed in a dryer for about 1 h until the temperature reached 20 °C. Successive weighing of the samples to constant weight followed.

Calculation of the percentage of ash

i.Average of successive weights [= Weight (capsule + sample)] − Capsule weight = x g ashii.(x g ash × 100)/original sample g = % ash

Three standard solutions of the following concentrations of 0.5 μg/mL, 1.2 μg/mL, and 10 μg/mL were prepared for each component (Cu, Ni, Co, Ca, Hg, Fe, Cr, Mg) using the working solutions by concentration 1 mg/mL. In addition, a HCl solution (10% *v*/*v*) was prepared, and 0.02 g of each sample was weighed. So, after adding 25 mL of preheated HCl solution was diluted to a final volume of 100 mL, each sample’s solution was measured on an atomic absorption spectrophotometer.

### 2.5. Antinutritional Evaluation

Trypsin inhibitor activity was determined according to the Kakade et al. [33] method.

For this method, the preparation of the following solutions was required.

Buffer Tris with pH = 8.2;Buffer Tris with pH = 7.5;Trypsin solution (5 mg trypsin enzyme was added in 0.001 M HCl solution in a final volume of 250 mL);Acetic acid solution 30% *v*/*v*;Substrate solution (40 mg BAPA was added to 1 mL DMSO and the volume was supplemented with Tris solution—pH = 8.2).

One g of bean powder was weighed and diluted with Tris solution (pH = 7.5). This was followed by stirring for 1 h and preparing the test tubes simultaneously in the following proportions: 2 mL water for the first test tube and 1.4 mL water and 0.6 mL bean powder solution for the second test tube. So, 2 mL of trypsin solution was added to all test tubes and placed in a water bath (35 °C). This was followed by the addition of a substrate solution (5 mL) and their retention in the water bath for 10 min. Afterward, 1 mL of the acetic acid solution (30% *v*/*v*) was introduced into each test tube to terminate the reaction. The samples were centrifuged for 15 min at 3500 rpm and measured by a double-beam spectrophotometer at a wavelength of 410 nm.

### 2.6. Statistical Analysis

Results are presented as the average ± standard deviation of three simultaneous assays. Comparison of means was conducted by least significance test (LSD) after Analysis of Variance (ANOVA), with SPSS 22.0 software (SPSS: Chicago, IL, USA), for one-factor randomized complete block design combined over locations. In order to quantify the observations mentioned in the manuscript, we proceeded to ANOVA calculations. For that purpose, namely, to quantify the differences between the groups and estimate the significance of these variances, two-way ANOVA was used. We used a confidence level of 95% and thus a value of a = 0.05. To quantify the differences between the groups and for estimation of significance, two-way ANOVA was used with a confidence level of 95%. 

## 3. Results and Discussion

The results obtained from the analysis of the seven landraces of beans regarding the nutritional factors, i.e., protein, lipid, dietary fiber, ash, and polyphenolic content, as well as the antinutritional factor trypsin inhibitor, are presented in Table 3. 

Protein contents ranged from 18.79 to 23.93 g/100 g DM. Genetic and environmental factors influence the protein concentration of bean varieties. Siddiq et al. [3] identified protein levels between 20.93 and 23.62 g/100 g DW in small red beans, kidney, and cranberry dry beans. Similar protein concentrations in different dry bean landraces and varieties are also reported in other studies in Greece and elsewhere [20,23,25,34,35,36,37,38,39]. In addition to genetic factors, protein values are influenced by genotype–environment interactions and also cultivation practices, as shown by different reports [20,25,38,39,40,41,42]. Bean lipids contribute to the ingestion of polyunsaturated fatty acids, which are highly related to a healthy diet [43]. The values of lipids ranged from 1.84 to 2.58 g/100 g DM, with no differences between the different environments and most of the genotypes. These values are comparable to those found in different bean genotypes evaluated in Greece [38,39] and other countries such as Japan [44], Nigeria [45], Brazil [36,41], and Turkey [23].

In the bean landraces evaluated in this study, starch contents varied from 40.14 to 55.26 g/100 g DM. These values are comparable to starch contents (39.43–51.92 g/100 g DM) found in Andean bean genotypes reported by Kajiwara et al. [36] and in different bean genotypes, landraces and cultivars grown in different locations [11,46]. Beans are an excellent source of micronutrients and have higher levels of minerals than other legumes and cereals [47]. In our study, the ash content of the seven bean landraces evaluated ranged from 4.31 to 5.2 g/100 g DM. Similar high ash levels have also been reported in other varieties and landraces of beans [11,35,48]. Furthermore, wild Mexican accessions of dry beans averaged a mineral content close to 4.57%, which is significantly higher than landraces, and improved cultivars, which averaged 3.84% and 4%, respectively [49]. Much lower levels of inorganic compounds (3.15–3.25 g/100 g DM) in different bean accessions growing for two years in two different environments in Portugal were also reported [37]. The fiber content in our landraces varied from 7.77 to 12.0 g/100 g DM. Mecha et al. [37] reported fiber contents for 121 dry bean accessions growing in different locations between 5.75 and 6.77 mg/100 g DM. In addition, fiber levels varied widely in wild accessions from 4.1% to 17% and in landraces and improved genotypes from 5.6% to 10.8% in a study from Mexico [49]. Phenolic compounds in beans may include a variety of flavonoids such as anthocyanins, flavonol, proanthocyanidins, tannins, and glycosides, as well as a wide range of phenolic acids [50]. In our study, the content of total phenolics in the bean landraces, which had white seed color in all the environments, ranged from 0.18 to 0.27 mg/g DM. Dinelli et al. [51] observed variations in the flavonoid content of Italian beans, ranging from 0.19 to 0.84 mg/g DM, and Campos-Vega et al. [52] reported total phenolics in white-seeded *Phaseolus vulgaris* between 0 and 0.4 mg/g DM from different studies. A very large variation in different phenolic compounds was reported for different dry bean extracts from seeds with different seed colors by Boateng et al. [53]. Trypsin inhibitors adversely affect protein digestibility, but health-promoting properties have also been evidenced [54]. A wide variation of trypsin inhibitors within common bean genotypes, as well as within each genotype in response to climatic conditions have been reported [8,55]. The levels found in the present study, 21.8–138.5 TIU/g DM, confirm the large variation in these compounds and are in agreement with those contents reported in dry bean cultivars and landraces in other studies [35,54,56].

### 3.1. Cultivation Area versus Nutritional and Antinutritional Content—Qualitative Study

Evaluation of the correlation between the variables revealed a strong correlation between protein and starch content (correlation coefficient 0.92). To explore the possibility of an underlying distribution of the area–year samples, a categorized scatter plot was created of the two variables, which are shown in Figure 2. The plot reveals a strong differentiation between one cultivation site against the other three. Specifically, all the genotypes cultivated in Agios Germanos (2015) had lower starch and protein values than the genotypes cultivated in Pili (2014), Pili (2015), and Patoulidio (2015). The Agios Germanos (2015) site was considered a low-to-intermediate-fertilized site and also had a slightly different soil type than the other locations used in this study (Table 2). The limited fertilization of the crop in this site could explain the lower levels of protein and starch in the bean seeds. Similarly, Slatni et al. [57] reported impaired seed quality as expressed by lower protein and soluble sugar contents in bean cultivars growing in soil with nutrient deficiencies. Low fertilization, especially the nitrogen one, can significantly affect the content of amino acids [58] and reduce the protein concentration in the seeds of common beans [59]. There was no other evident correlation between other variables. However, increasing phenol concentration in the sample lowered other components except for trypsin. Similarly, ash content was negatively correlated with all except starch and protein (Table 4).

Fiber, ash, phenol, and trypsin inhibitor content appeared to be affected by crop year, as there are significant differences in these variables’ content in the samples from Pili between the years 2014 and 2015. Phenol content was significantly affected by crop year, giving similar results in cultivations of the same year (Pili 2015, Patoulidio 2015, Agios Germanos 2015) and significantly different results for the year 2014 (Pili 2014). Different environmental conditions can affect the phenolic content of dried beans [51,60]. Trypsin inhibitor proteins appear to be the most significantly affected variable from the cultivation conditions, thus better discriminating cultivations areas and crop years. In Pili 2014, the TIU levels for all the landraces tested ranged from 54.2–74.50, in Agios Germanos 2015 between 21.8 and 74.80, whereas in Pili 2015 and in Patoulidio 2015, there was a significant increase in the concentrations of trypsin inhibitors in almost all the landraces evaluated with levels ranging from 46.6 to 99.33 and 88.40 to 138.50 TIU/g DM, respectively. The areas of Pili and Patoulidio in the year 2015, during the beginning of flowering (late June), suffered a catastrophic hail with a greater impact on the Patoulidio area. This severely distressed the plants, which recovered later in the cultivation period. Full anthesis, seed filling, and seed ripening period were delayed resulting in additional stress due to exposure to high temperatures. Similarly, stressful conditions during the vegetative and reproductive stages of a bean crop have been shown to significantly increase trypsin inhibitors’ expression [8,35]. In contrast, lipid content appeared not to be affected by the different soil types, years, and cultivation conditions.

Plotting phenols content versus starch content discriminated groups in terms of crop year and different cultivation conditions (Figure 3). Strong discrimination of the Agios Germanos site in terms of phenolic and starch content was seen as compared to other sites. Moreover, the cultivation year of the same location was found to influence the phenolic and starch content, as in the case of the Pili 2014 and 2015 crops. Moreover, the Pili 2015 crop starch and phenolic content were similar to the Patoulidio location (Figure 3).

Overall better nutritional value in terms of protein and starch content was attributed to the cultivation areas Pili and Patoulidio, with no significant differences in other nutritional variables (lipid, phenolic content, ash), whereas Patoulidio appears to have higher trypsin inhibitor proteins (though higher fiber content).

### 3.2. Genotype of Locally Adapted Beans versus Nutritional and Antinutritional Content Qualitative Study

Box plots did not reveal significant differences between the genotypes regarding protein and starch content (Figure 4a,b), which, as mentioned earlier, were more affected by the site of cultivation. Lipid content, which was not significantly affected by the different environments, appeared to be higher with lower altitude genotypes (Figure 4c). The same trend was shown for fiber content (higher fiber content in lower altitude genotypes) (Figure 4d). Phenolic content showed no significant difference between the different genotypes (Figure 4e). Differences in phenolic composition between varieties could be related to the color of the seed. Detailed studies suggest that variability in phenolic content is more due to genotype than seed coat color [61]. However, this was not the case in our study. Ash content seemed to be higher in the genotypes that originated from higher altitudes (Figure 4f). Further conclusions could be drawn from the mathematical model due to the strong effect of other factors, e.g., cultivation area and crop year.

To quantify the differences between the groups and for estimation of significance, two-way ANOVA was used with a confidence level of 95% (Table 5). In Table 5, we present the Fisher values and probability error for the nutritional and antinutritional variables of the seven locally adapted genotypes of beans in four cultivation conditions.

All variables were significantly affected by the environment except lipid content, while no significant effect of genotype was seen on any of the variables (Table 5). Among these, protein, starch, and phenol content were highly affected by agronomical practices compared to ash and fiber content.

### 3.3. Nutritional and Antinutritional Quality of the Different Genotypes

Since there were no significant differences in protein, starch, lipid, and phenolic content between the different genotypes, we considered in terms of trypsin inhibitor, fiber, and ash content, in order to quantify the nutritional and antinutritional quality of the genotypes. However, there are some differences that are significant if we group some of them based on previous studies.

### 3.4. Discriminant Analysis

Due to the fact that various different factors affected the nutritional and antinutritional content, discriminant analysis was used in order to isolate each factor. In order to discriminate cultivation conditions, we used the variables starch, phenol, trypsin inhibitor, fiber, and ash content. Protein content was not used due to the significant correlation with starch, whereas the results of the Two-Way ANOVA restrained the use of the lipid content (as described earlier). The results of the analysis provided three roots, the contribution of which are presented in Table 6. The four environments are satisfactorily discriminated.

The first discriminant function is weighted most heavily by the starch and phenolic content. The other three variables also contribute to this function. The second function seems to be marked mostly by variables: fiber and lipid and, to a lesser extent, by the other three variables. The third function is affected mostly by trypsin inhibitor proteins and fiber content. In Table 6, the explained variance accounted for each function is shown. As can be seen from the eigenvalues, the first function accounts for 65% of the explained variance, while the second and third functions account for 26% and 9%, respectively.

The first discriminant function discriminates mostly between Agios Germanos and the other agronomic environments. The canonical mean of Agios Germanos is quite different from that of the other groups, thus discriminating. The second discriminant function seems to distinguish mainly between Pili 2015 and the other cultivation sites. However, based on the review of the eigenvalues earlier, the magnitude of the discrimination is much smaller.

The scatter plot of canonical scores for function 1 and function 2 plots confirmed the interpretation so far. The cultivation conditions on Pili 2014 and Agios Germanos are plotted much further than the Patoulidio and Pili 2015 sites (Figure 5a). Thus, the first function discriminates between starch and phenolic content. The second function seems to provide some discrimination between fiber and lipid content.

The scatterplot of canonical scores for function 1 and function 3 plots confirmed the similarities between Pili 2014 and Pili 2015 (Figure 5b), while Patoulidio and Agios Germanos are excellently distinguished.

### 3.5. Mathematical Model

In order to trace and quantify the impact of genotype on nutritional and antinutritional variables, we developed a simplified model simultaneously for both factors (cultivation area–genotype) to separate the impact of cultivation area and genotypes in the experimental results. The model used was:y = C a_i_ b_j_
where C is a constant that corresponds to the average of each variable (protein starch, etc.), a_i_ is the impact coefficient for each cultivation condition (i = 1, 2, 3, 4), and b_j_ is the impact coefficient for each genotype (j = 1 to 7 for the genotypes A, B, C, D, E, F, and G, respectively). By this method one can isolate each factor and study separately their effect on the experimental results (Table 7).

The model provided satisfactory results (taking into account the estimated variance) except for trypsin inhibitor proteins. On trypsin inhibitors, protein content appears to contribute a second-order effect that cannot be reliably (confidently) estimated. The trypsin inhibitor data will be analyzed qualitatively.

The mathematical model suggests a lesser protein and starch content of Agios Germanos (>0.95) and a similar impact of the three remaining cultivation conditions (~1,1), confirming the previous analysis. Moreover, the model indicates a similar impact of all the cultivation areas–crop year on lipid content. Fiber content appears to be significantly lower for Pili 2015, while the other three cultivation conditions seem to have the same effect on its content (~1,1). Phenolic content appears to be significantly lower for Pili 2014, and this is already attributed to different environmental conditions in that year. Ash content is significantly different only between the cultivation areas Pili 2014 and the two cultivation areas Pili 2015 and Agios Germanos.

### 3.6. Nutritional Evaluation versus Genotype, Based on Mathematical Model

Finally, the nutritional evaluation of each genotype can be analyzed based on the mathematical model. Genotype “E” showed higher lipid and fiber content, while genotype “D” showed higher protein, starch, phenolic, and ash content. On the other hand, the protein was found to be lower in genotype “G”, lipid and fiber in genotype “D”, starch in genotype “B”, phenol in genotype “C”, ash in genotype “F”, and trypsin inhibitor protein in genotype “B” (Figure 6).

## 4. Conclusions

The effect of genotype, agronomic conditions (soil type, fertilization, etc.), and environment (microclimate, specific weather conditions) on the nutritional and anti-nutritive properties of locally adapted Phaseolus vulgaris races was the subject of this study.

From the overall analysis, protein and starch content were mainly affected by agronomical conditions, namely soil type, irrigation, and fertilization practices, and marginally affected by genotype and weather conditions, whereas lipid and ash contents on dry beans were less affected by agronomical conditions. Thus, such compounds may be used as a suitable index for genotype discrimination. Fiber appeared to be affected by both genotype and agronomical conditions, while phenolic and trypsin inhibitor contents were considerably affected by stress conditions either in spotted points or during the whole cultivation period, distressing mostly the crop. Due to the organic cultivation methods applied, concentrations of toxic metals were only determined in traces that did not exceed established limits.

## Figures and Tables

**Figure 1 foods-12-01119-f001:**
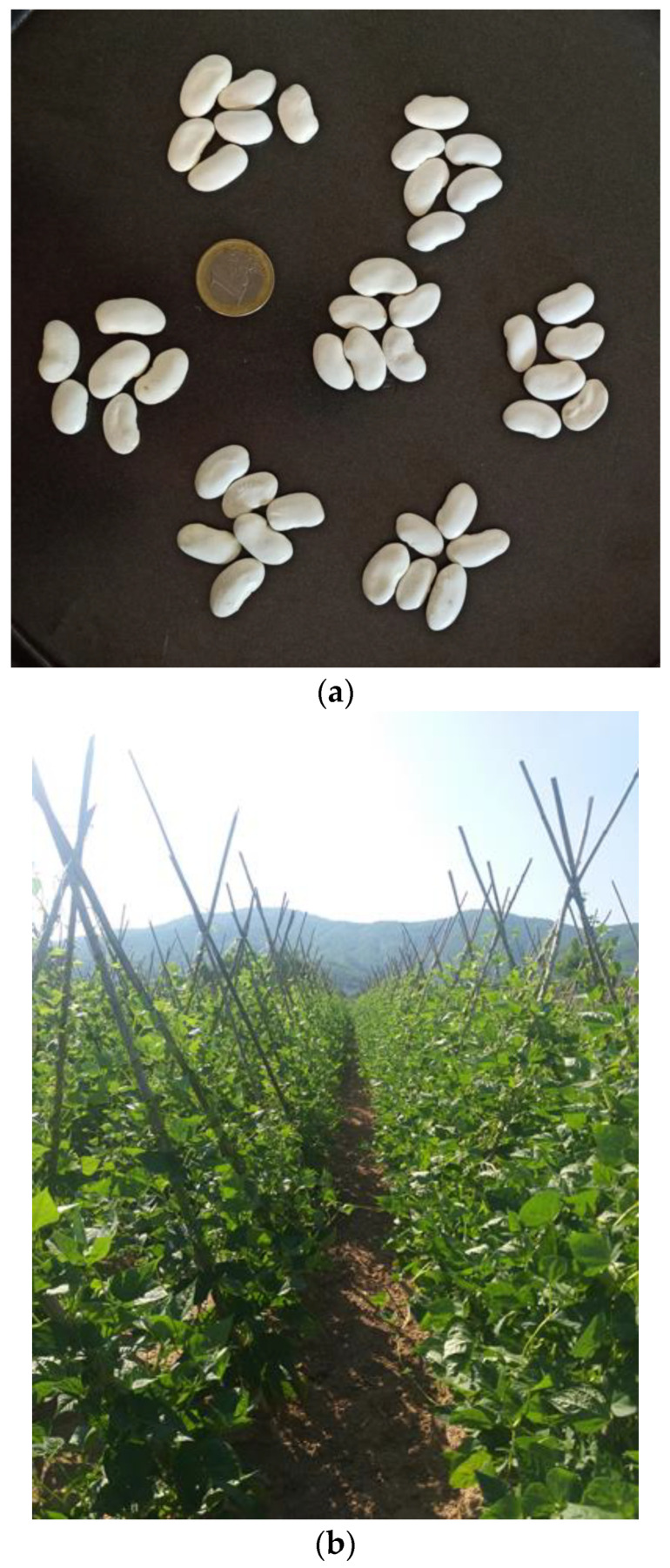
(**a**): White beans. (**b**): *Phaseolus vulgaris*.

**Figure 2 foods-12-01119-f002:**
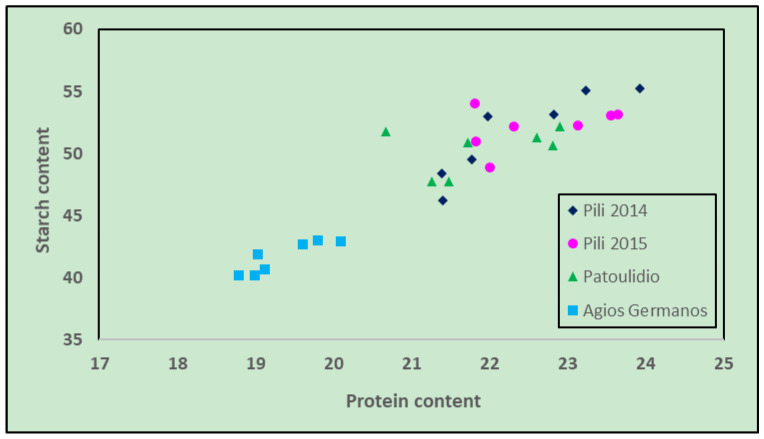
Scatter plot of protein content (g/100 g dry matter) versus starch content (g/100 g dry matter).

**Figure 3 foods-12-01119-f003:**
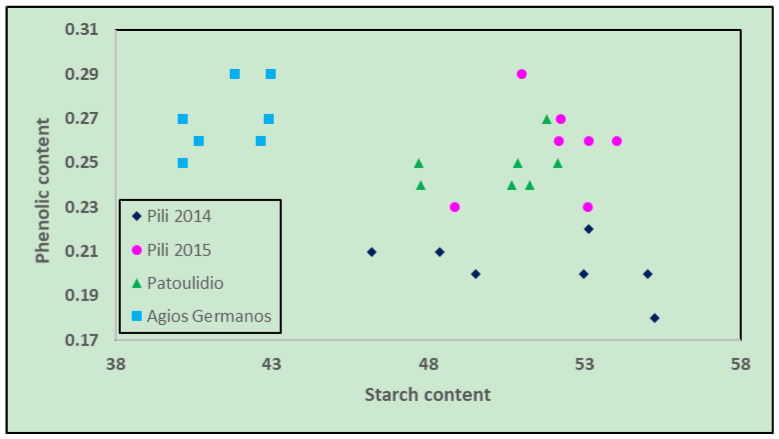
Chart of phenol content versus Starch content.

**Figure 4 foods-12-01119-f004:**
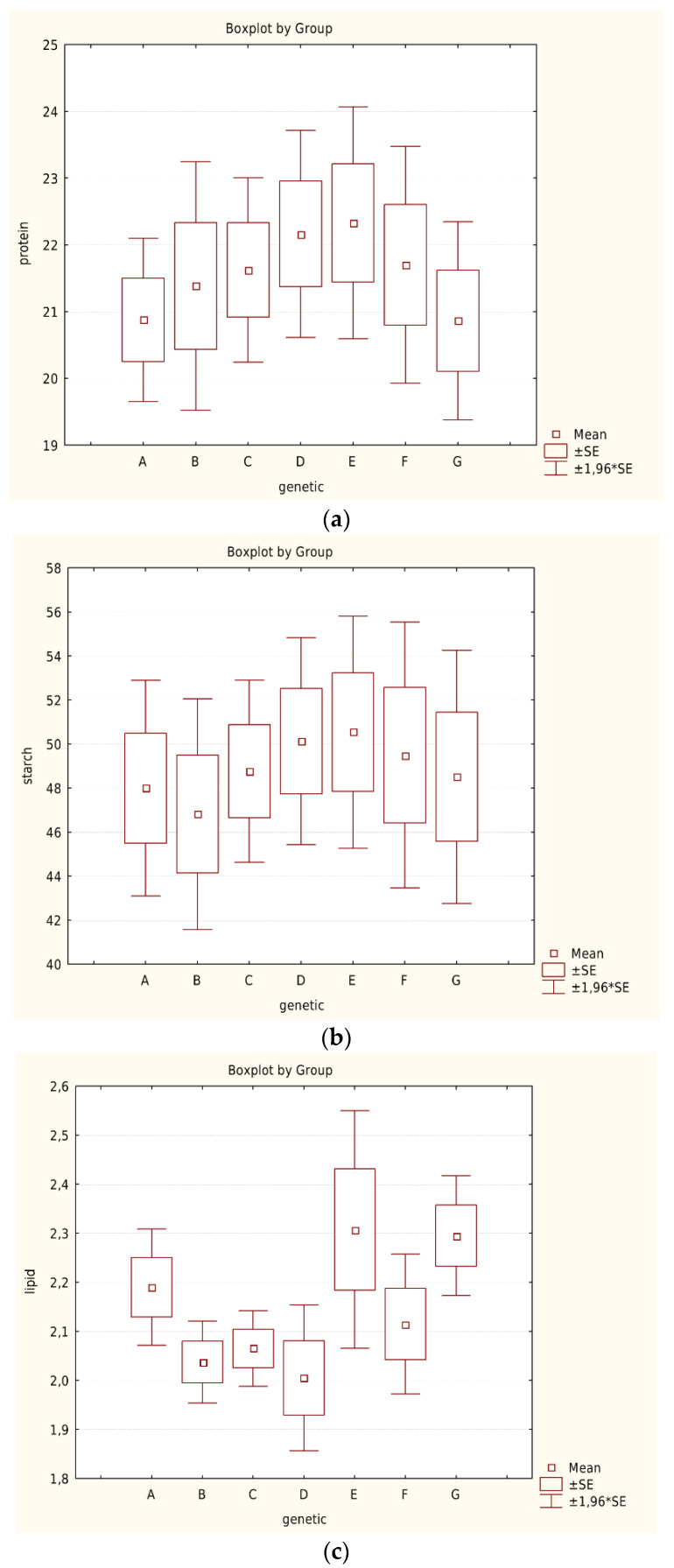
(**a**): Protein content of the different genotypes. (**b**): Starch content of the different genotypes. (**c**): Lipid content of the different genotypes. (**d**): Fiber content of the different genotypes. (**e**): Phenol content of the different genotypes. (**f**): Ash content of the different genotypes.

**Figure 5 foods-12-01119-f005:**
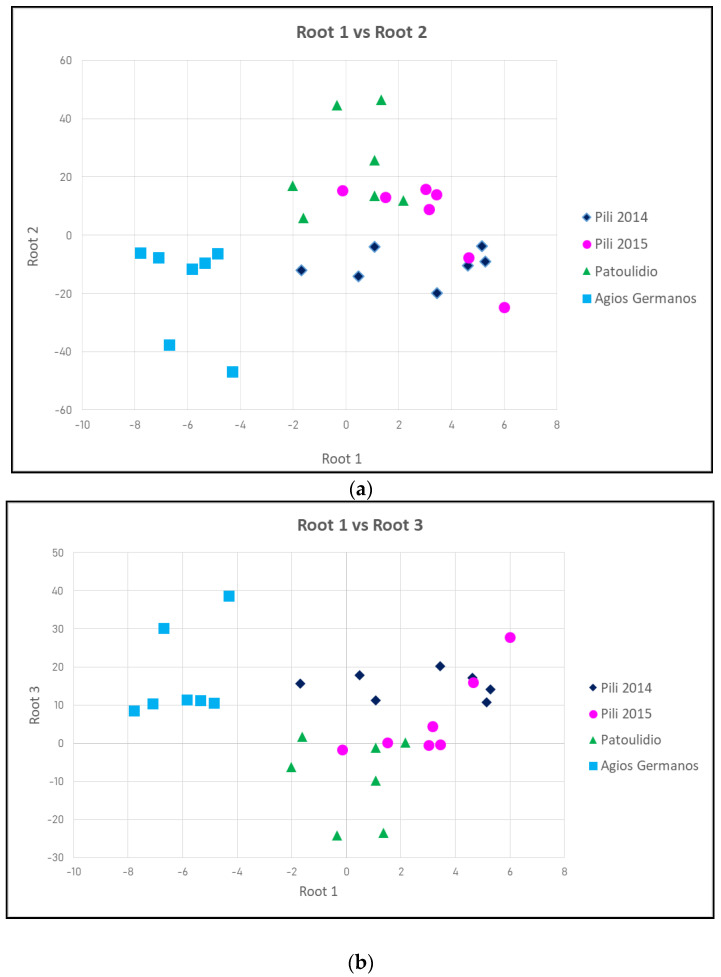
(**a**): Scatter plot of canonical scores for function 1 and function 2 plots. (**b**): Scatterplot of canonical scores for function 1 and function 3 plots.

**Figure 6 foods-12-01119-f006:**
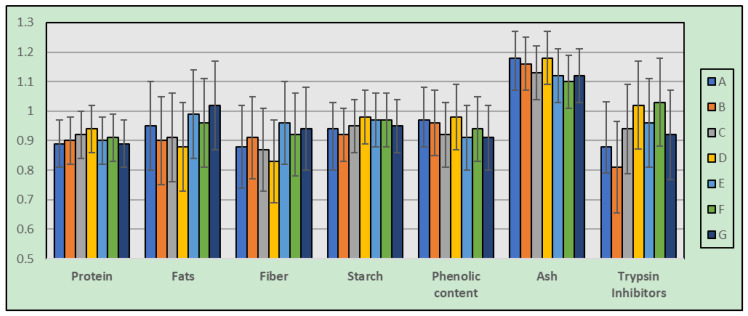
Nutritional evaluation of each genotype analyzed based on the mathematical model (error bars represent standard deviations, *n* = 3).

**Table 1 foods-12-01119-t001:** Collection place, growth habit, seed shape, coat veining, and brilliance of the seven dry bean landraces.

Landrace	Collecting Place	Growth Habit *	Shape	Seed Color	Veining	Brilliance
A	Agios Germanos	IV	Kidney	White	-	Medium
B	Plati	IV	Kidney	White	-	Medium
C	Nakolets	IV	Kidney	White	-	Shiny
D	Laimos	IV	Oval	White	-	Shiny
E	Chrisoupoli	IV	Kidney	White	-	Mat/medium
F	Kastoria	IV	Kidney	White	-	Shiny
G	Florina	IV	Kidney	White	+	Medium

* IV indeterminate climbing type.

**Table 2 foods-12-01119-t002:** Soil physicochemical properties in the different cultivation areas.

Site	Year	Soil Type	pH	CaCO_3_ (%)	EC (mS/cm)	Organic Matter (%)
Pili	2014	L	7.56	4.8	0.264	1.15
Pili	2015	L	7.56	4.8	0.380	2.37
Patoulideio	2015	L	6.75	0	0.237	1.11
Agios Germanos	2015	SL	6.32	0	0.253	1.37

L = loam soil, SL = sandy loam soil.

**Table 3 foods-12-01119-t003:** Estimation of nutritional and antinutritional components in dry beans.

Origin	Genotype	Year	Proteins *	Lipids *	Dietary Fibers *	Ash *	Starch *	Total Phenolics **	Trypsin Inhibitor ***
Pili	A	2014	21.39 ± 0.5	2.11 ± 0.32	8.90 ± 0.56	5.20 ± 0.57	48.38 ± 0.38	0.21 ± 0.01	61.90
Pili	B	2014	21.41 ± 0.44	1.95 ± 0.25	9.11 ± 0.47	4.87 ± 0.48	46.19 ± 0.41	0.21 ± 0.03	65.20
Pili	C	2014	21.77 ± 0.37	2.15 ± 0.15	8.89 ± 0.45	5.01 ± 0.53	49.53 ± 0.39	0.20 ± 0.02	74.50
Pili	D	2014	22.83 ± 0.51	1.84 ± 0.17	8.61 ± 0.51	5.10 ± 0.55	53.16 ± 0.26	0.22 ± 0.03	65.20
Pili	E	2014	23.93 ± 0.71	2.58 ± 0.11	11.24 ± 0.32	4.64 ± 0.44	55.26 ± 0.21	0.18 ± 0.00	67.40
Pili	F	2014	23.24 ± 0.39	2.31 ± 0.29	10.69 ± 0.39	4.87 ± 0.49	55.04 ± 0.22	0.20 ± 0.02	74.10
Pili	G	2014	21.98 ± 0.42	2.46 ± 0.36	12 ± 0.27	4.79 ± 0.46	53 ± 0.24	0.20 ± 0.02	54.20
Pili	A	2015	21.82 ± 0.45	2.33 ± 0.19	8.34 ± 0.53	4.93 ± 0.51	54.05 ± 0.25	0.26 ± 0.02	46.60
Pili	B	2015	23.65 ± 0.38	2.05 ± 0.22	8.05 ± 0.54	4.86 ± 0.46	53.15 ± 0.29	0.26 ± 0.01	68.60
Pili	C	2015	22.31 ± 0.54	2.02 ± 0.24	7.95 ± 0.59	4.54 ± 0.41	52.20 ± 0.31	0.26 ± 0.01	90.00
Pili	D	2015	23.13 ± 0.46	2.09 ± 0.35	7.77 ± 0.49	4.72 ± 0.43	52.24 ± 0.39	0.27 ± 0.04	98.90
Pili	E	2015	23.56 ± 0.46	2.09 ± 0.2	8.93 ± 0.34	4.84 ± 0.47	53.10 ± 0.33	0.23 ± 0.02	96.80
Pili	F	2015	21.83 ± 0.52	2.13 ± 0.13	8.85 ± 0.41	4.31 ± 0.34	50.99 ± 0.2	0.29 ± 0.07	95.88
Pili	G	2015	22.00 ± 0.37	2.16 ± 0.28	8.29 ± 0.44	4.49 ± 0.39	48.87 ± 0.37	0.23 ± 0.01	99.33
Patoulidio	A	2015	21.26 ± 0.51	2.25 ± 0.21	10.60 ± 0.38	5.19 ± 0.56	47.71 ± 0.33	0.25 ± 0.02	102.40
Patoulidio	B	2015	21.48 ± 0.36	2.15 ± 0.29	10.60 ± 0.37	4.78 ± 0.46	47.78 ± 0.34	0.24 ± 0.01	88.40
Patoulidio	C	2015	22.81 ± 0.63	1.98 ± 0.41	9.39 ± 0.42	4.55 ± 0.41	50.68 ± 0.28	0.24 ± 0.01	96.90
Patoulidio	D	2015	22.90 ± 0.65	2.17 ± 0.31	8.96 ± 0.43	4.93 ± 0.50	52.15 ± 0.25	0.25 ± 0.01	138.50
Patoulidio	E	2015	21.73 ± 0.58	2.02 ± 0.34	10.47 ± 0.29	4.70 ± 0.42	50.88 ± 0.29	0.25 ± 0.02	137.00
Patoulidio	F	2015	22.61 ± 0.49	2.05 ± 0.4	9.57 ± 0.28	4.69 ± 0.46	51.27 ± 0.39	0.24 ± 0.01	112.30
Patoulidio	G	2015	20.68 ± 0.39	2.30 ± 0.15	9.40 ± 0.27	4.87 ± 0.49	51.81 ± 0.35	0.27 ± 0.02	94.60
Agios Germanos	A	2015	19.04 ± 0.3	2.07 ± 0.32	9.32 ± 0.33	4.86 ± 0.48	41.83 ± 0.42	0.29 ± 0.01	21.80
Agios Germanos	B	2015	19.00 ± 0.26	2.00 ± 0.28	10.64 ± 0.25	4.70 ± 0.42	40.15 ± 0.45	0.27 ± 0.06	34.60
Agios Germanos	C	2015	19.61 ± 0.29	2.11 ± 0.29	10.30 ± 0.28	5.05 ± 0.52	42.66 ± 0.49	0.26 ± 0.01	69.90
Agios Germanos	D	2015	19.80 ± 0.35	1.92 ± 0.36	9.23 ± 0.37	4.76 ± 0.45	42.97 ± 0.47	0.29 ± 0.01	73.50
Agios Germanos	E	2015	20.10 ± 0.4	2.20 ± 0.16	11.98 ± 0.22	4.49 ± 0.29	42.92 ± 0.35	0.27 ± 0.01	67.50
Agios Germanos	F	2015	19.13 ± 0.21	1.97 ± 0.19	9.81 ± 0.31	4.42 ± 0.38	40.68 ± 0.51	0.26 ± 0.01	72.40
Agios Germanos	G	2015	18.79 ± 0.27	2.26 ± 0.21	10.27 ± 0.24	4.54 ± 0.40	40.14 ± 0.55	0.25 ± 0.01	74.80

* g/100 g dry matter; ** mg/g equivalents of gallic acid in dry matter; *** TIU/g dry matter.

**Table 4 foods-12-01119-t004:** Correlation analysis between nutritional and antinutritional variables.

	Protein	Lipid	Fiber	Starch	Phenol	Ash	Trypsin
Protein	1.00	0.19	−0.27	0.92	−0.53	0.36	0.42
Lipid	0.19	1.00	0.55	0.30	−0.36	−0.09	−0.09
Fiber	−0.27	0.55	1.00	−0.26	−0.15	−0.15	−0.32
Starch	0.92	0.30	−0.26	1.00	−0.50	0.33	0.40
Phenol	−0.53	−0.36	−0.15	−0.50	1.00	−0.37	0.20
Ash	0.36	−0.09	−0.15	0.33	−0.37	1.00	−0.01
Trypsin	0.42	−0.09	−0.32	0.40	0.20	−0.01	1.00

**Table 5 foods-12-01119-t005:** Fisher values and probability error for the nutritional and antinutritional variables of the seven locally adapted bean genotypes in four cultivation conditions.

Effect	Genotype				Agronomical-Year			
Variable	df	F	*p*	Fc	df	F	*p*	Fc
Proteins	6	0.21	9.68 × 10^−1^	2.66	3	26.41	8.90 × 10^−8^	3.15
Lipid	6	1.67	1.77 × 10^−1^	2.66	3	0.89	4.61 × 10^−1^	3.15
Fibber	6	0.64	6.90 × 10^−1^	2.66	3	9.24	3.04 × 10^−4^	3.15
Starch	6	0.15	9.86 × 10^−1^	2.66	3	36.68	4 × 10^−9^	3.15
Phenol	6	0.23	9.62 × 10^−1^	2.66	3	30.36	2.45 × 10^−8^	3.15
Ash	6	1.39	2.65 × 10^−1^	2.66	3	3.28	3.82 × 10^−2^	3.15
Trypsin inhibitor	6	0.58	7.39 × 10^−1^	2.66	3	22.11	4.33 × 10^−7^	3.15

**Table 6 foods-12-01119-t006:** Standardized coefficients for canonical variables.

	Root 1	Root 2	Root 3
Starch	0.86645	0.205428	0.173549
Phenol	−0.71622	0.495221	0.070624
Trypsin	−0.02678	0.782007	−0.553807
Fiber	−0.38564	−0.247337	−0.948749
Ash	0.21825	0.025805	−0.716898
Eigenval	10.19219	4.125158	1.414211
Cum. Prop	0.64788	0.910104	1.000000

**Table 7 foods-12-01119-t007:** Nutritional evaluation versus cultivation conditions based on the mathematical model.

Cultivation Condition		Protein	Lipid	Fiber	Starch	Phenol	Ash	Trypsin Inh
Pili 2014	a1	1.16 ± 0.07	1.11 ± 0.12	1.16 ± 0.13	1.11 ± 0.07	0.88 ± 0.09	0.90 ± 0.05	0.88 ± 0.132
Pili 2015	a2	1.15 ± 0.07	1.03 ± 0.12	0.95 ± 0.13	1.11 ± 0.07	1.11 ± 0.09	0.87 ± 0.05	1.21 ± 0.124
Patoulidio	a3	1.14 ± 0.07	1.05 ± 0.12	1.15 ± 0.13	1.08 ± 0.07	1.07 ± 0.09	0.89 ± 0.05	1.28 ± 0.122
Agios Germanos	a4	0.97 ± 0.07	1.04 ± 0.12	1.18 ± 0.13	0.89 ± 0.07	1.18 ± 0.09	0.84 ± 0.05	0.89 ± 0.132
Genotype								
A	b1	0.89 ± 0.08	0.95 ± 0.15	0.88 ± 0.14	0.94 ± 0.09	0.97 ± 0.11	1.18 ± 0.09	0.88 ± 0.152
B	b2	0.90 ± 0.08	0.90 ± 0.15	0.91 ± 0.14	0.92 ± 0.09	0.96 ± 0.11	1.16 ± 0.09	0.81 ± 0.154
C	b3	0.92 ± 0.08	0.91 ± 0.15	0.87 ± 0.14	0.95 ± 0.09	0.92 ± 0.11	1.13 ± 0.09	0.94 ± 0.151
D	b4	0.94 ± 0.08	0.88 ± 0.15	0.83 ± 0.14	0.98 ± 0.09	0.98 ± 0.11	1.18 ± 0.09	1.02 ± 0.149
E	b5	0.90 ± 0.08	0.99 ± 0.15	0.96 ± 0.14	0.97 ± 0.09	0.91 ± 0.11	1.12 ± 0.09	0.96 ± 0.151
F	b6	0.91 ± 0.08	0.96 ± 0.15	0.92 ± 0.14	0.97 ± 0.09	0.94 ± 0.11	1.10 ± 0.09	1.03 ± 0.149
G	b7	0.89 ± 0.08	1.02 ± 0.15	0.94 ± 0.14	0.95 ± 0.09	0.91 ± 0.11	1.12 ± 0.09	0.92 ± 0.151
	χ2	14.45	0.49	9.65	89.27	0.0034	0.61	922.92
Est	Var	0.85	0.03	0.57	5.25	0.0002	0.04	54.29

## Data Availability

Data is contained within the article.

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
