# Peer review of "Microclimate and Genotype Impact on Nutritional and Antinutritional Quality of Locally Adapted Landraces of Common Bean (Phaseolus vulgaris L.)"

_foods, 2023, doi:10.3390/foods12061119_

Round 1

Reviewer 1 Report

The manuscript reported a comprehensive analysis of microclimatic effect and grain quality related phenotype of large-seeded bean landraces from Northern Greece and North Macedonia.  The results provide insight for the understanding the locally adaption of Phaseolus vulgaris landraces under different agroecological environments. The relevant quality assessment of genetic resources is relatively interesting for molecular breeders of large-seeded bean. The study was presented and the conclusions were supported by multiple evidences. The figures are in sound quality.

In general, the manuscript should be considering the comments:

1.  The overall scientific name should be italic, please check it carefully.

2.   The “Results” may be changed as “Results and Discussion”.

3.  Figure 1 to 3, and Table 1 were not cited in the main text, please check it.

4.  The Figure 1 and 2 an be merged as one figure, and the scale for the Fig.2 will be indicated.

   5. Table 3 is not essential, which can be described in the text.

  6. Overall statistic analysis for significance of the Tables needs to be further indicated.

   7. It will be interesting to have the correlation analysis of Table 1 and Table 4. to reveal the effect of local soil parameters to the grain components.

Author Response

The manuscript reported a comprehensive analysis of microclimatic effect and grain quality related phenotype of large-seeded bean landraces from Northern Greece and North Macedonia.  The results provide insight for the understanding the locally adaption of Phaseolus vulgaris landraces under different agroecological environments. The relevant quality assessment of genetic resources is relatively interesting for molecular breeders of large-seeded bean. The study was presented and the conclusions were supported by multiple evidences. The figures are in sound quality.

-We would like to thank the reviewer for his/her comments.

In general, the manuscript should be considering the comments:

  1. The overall scientific name should be italic, please check it carefully.

-The scientific name has been corrected to italic throughout the manuscript as suggested.

  1. The “Results” may be changed as “Results and Discussion”.

- Thank you for the correction. The section has been changed to “Results and Discussion”.

  1. Figure 1 to 3, and Table 1 were not cited in the main text, please check it.

- Thank you for the correction. All Figures and Tables are now cited in the main text, as suggested.

  1. The Figure 1 and 2 an be merged as one figure, and the scale for the Fig.2 will be indicated.

-These figures have been merged to Figure 1a and Figure 1b, as suggested.

  1. Table 3 is not essential, which can be described in the text.

-Table 3 has been deleted.

  1. Overall statistic analysis for significance of the Tables needs to be further indicated.

-In order to quantify the observations mentioned in the manuscript, we proceeded to ANOVA calculations. For that purpose, namely to quantify the differences between the groups and estimate the significance of these variances, Two Way ANOVA was used. We used a confidence level of 95% and thus a value of a= 0.05. The F values for each variable are presented in table 5. To quantify the differences between the groups and for estimation of significance, two way ANOVA was used with confidence level of 95%. In the Table 5 we present the Fisher values and probability error for the nutritional and antinutritional variables of the seven locally adapted genotypes of bean in four cultivation conditions.

Reviewer 2 Report

The manuscript entitled “Microclimate and genotype impact on nutritional and anti-nutritional quality of locally adapted landraces of common bean (Phaseolus vulgaris L.)” aimed to assess the impact of genotype, location, and type of cultivation on the nutrient and anti-nutrient components of seven large-seeded common bean. The manuscript is useful and provides new resources for utilizing landraces of crops.

However, I have some concerns attached below.

1. “Phaseolus vulgaris” in the title and the 2nd line of Abstract should be italic.

2. The authors insist that the limitation of the use of the cultivars is due to the diverse compounds in their grain. How did they reach the conclusion? Since the contents of the compounds in the commercial cultivars have not been detected.

3. In Line 4 in Abstract: refers to Pyli, but Pili after that. Please confirm it.

4. Line 9 in the Introduction part: represents?

5. Statistical analysis of the generated data has not been detected. In my opinion, it is useful for their conclusion.

6. The vertical lines in each column in Figure 7 look like the same for the same indicator. Please confirm that.

Author Response

The manuscript entitled “Microclimate and genotype impact on nutritional and anti-nutritional quality of locally adapted landraces of common bean (Phaseolus vulgaris L.)” aimed to assess the impact of genotype, location, and type of cultivation on the nutrient and anti-nutrient components of seven large-seeded common bean. The manuscript is useful and provides new resources for utilizing landraces of crops.

-We would like to thank the reviewer for his/her comments.

However, I have some concerns attached below.

  1. “Phaseolus vulgaris” in the title and the 2nd line of Abstract should be italic.

-Thank you for the correction. This has been corrected to “Phaseolus vulgaris L.”

  1. The authors insist that the limitation of the use of the cultivars is due to the diverse compounds in their grain. How did they reach the conclusion? Since the contents of the compounds in the commercial cultivars have not been detected.

-Possible explanations on the variable composition may be due to geographic, climatic or cultivar reasons. As geographic variation is minimal, climatic conditions are similar the area in West part of Macedonia, main differences should be due to different cultivars/strains. However, further more focused experimentations are necessary to come to certain conclusion.

  1. In Line 4 in Abstract: refers to Pyli, but Pili after that. Please confirm it.

 -Thank you for the correction. This has been corrected to “Pili”.

  1. Line 9 in the Introduction part: represents?

 -This has been corrected to “represents”. Thank you.

  1. Statistical analysis of the generated data has not been detected. In my opinion, it is useful for their conclusion.

-In order to quantify the observations mentioned in the manuscript, we proceeded to ANOVA calculations. For that purpose, namely to quantify the differences between the groups and estimate the significance of these variances, Two Way ANOVA was used. We used a confidence level of 95% and thus a value of a= 0.05. The F values for each variable are presented in table 5. To quantify the differences between the groups and for estimation of significance, two way ANOVA was used with confidence level of 95%. In the Table 5 we present the Fisher values and probability error for the nutritional and antinutritional variables of the seven locally adapted genotypes of bean in four cultivation conditions.

  1. The vertical lines in each column in Figure 7 look like the same for the same indicator. Please confirm that.

-The text that describes this Figure has been rewritten in order to better explain the Figure.

Reviewer 3 Report

This is an interesting resarch work, which discusses the effect of genotype, agronomical conditions, and environmental differences in nutritional and antinutritional properties of Phaseolus vulgaris L.

A few suggestions to improve readability and scientific soundness of the content are given in the following.

The use of preposition, punctuation and grammar is weak.

The use of scientific names has not been uniform. 

The text requires a thorough recheck for readability and remove minor mistakes in the text. I suggest the authors to have a similarity check of the text if they have not done it earlier.

Sowing of seeds in field is not clearly mentioned (agronomic conditions like fertilizer application, irrigation).

Use SI symbols / units such as for Celsius.

Subtitles are not uniform.

Abbreviations if used, should follow a uniform pattern throughout the text.

The statistical analysis description is not clear (rewrite it).

Place the tables on their required positions especially the Table 5.

Add “Discussion” title with results (Results and Discussion) or write discussion separately.

The results were mainly compared with studies conducted outside the Greece.

Write descriptions for the Figure 5 series.

Elaborate the conclusions as compared to your article title.

Author Response

This is an interesting resarch work, which discusses the effect of genotype, agronomical conditions, and environmental differences in nutritional and antinutritional properties of Phaseolus vulgaris L.

-We would like to thank the reviewer for his/her comments.

A few suggestions to improve readability and scientific soundness of the content are given in the following.

  1. The use of preposition, punctuation and grammar is weak.

- Thank you for the correction. The prepositions, punctuation and grammar have been corrected throughout the manuscript.

  1. The use of scientific names has not been uniform. 

- Thank you for the correction. The use of scientific names has been corrected throughout the manuscript.

  1. The text requires a thorough recheck for readability and remove minor mistakes in the text. I suggest the authors to have a similarity check of the text if they have not done it earlier.

-Thank you for the correction. After proofreading mistakes have been corrected throughout the manuscript.

  1. Sowing of seeds in field is not clearly mentioned (agronomic conditions like fertilizer application, irrigation).

The agronomic conditions are described in Part 2.2 Experimental design. Specifically:

-All experiments were set-up on fully organic-certified fields. Pili was considered a well-irrigated and fully fertilized site whereas Patoulidio and Agios Germanos were considered intermediate irrigated and fertilized sites due to scarcity of irrigation water during the maturation period of the bean crop.

Regarding Fertilization: Fertilization in Pili was performed every year while in Patoulidio and Agios Germanos every two years using 50 ton per hectare of dry cattle manure in all fields. The agronomic practices in all sites followed those of organic agriculture recommended for the crop in this region. Regarding fertilization is clear in the text: "Fertilization in Pili was performed every year while in Patoulidio and Agios Germanos every two years using 50 ton per hectare of dry cattle manure in all fields.

Regarding irrigation: "Pili was considered a well-irrigated and fully fertilized site whereas Patoulidio and Agios Germanos were considered intermediate irrigated and fertilized sites due to scarcity of irrigation water during the maturation period of the bean crop. Irrigation in Pili site was performed once a week with a drip irrigation system according to the needs of the crop up until physiological maturity of the crop whereas in Patoulidio and Agios Germanos irrigation was performed with furrow flooding once a week and just before and during the seed filling stage every two weeks due to scarcity of irrigation water.

  1. Use SI symbols / units such as for Celsius.

-Thank you.  This has been corrected throughout the manuscript.

  1. Subtitles are not uniform.

-This has been corrected throughout the manuscript.

  1. Abbreviations if used, should follow a uniform pattern throughout the text.

-This has been corrected throughout the manuscript.

  1. The statistical analysis description is not clear (rewrite it).

-In order to quantify the observations mentioned in the manuscript, we proceeded to ANOVA calculations. For that purpose, namely to quantify the differences between the groups and estimate the significance of these variances, Two Way ANOVA was used. We used a confidence level of 95% and thus a value of a= 0.05. The F values for each variable are presented in table 5. To quantify the differences between the groups and for estimation of significance, two way ANOVA was used with confidence level of 95%. In the Table 5 we present the Fisher values and probability error for the nutritional and antinutritional variables of the seven locally adapted genotypes of bean in four cultivation conditions.

  1. Place the tables on their required positions especially the Table 5.

-All tables have been placed on their required positions.

  1. Add “Discussion” title with results (Results and Discussion) or write discussion separately.

-This has been changed to “Results and Discussion”.

  1. The results were mainly compared with studies conducted outside the Greece.

 -Although there is no combined study about nutritional and antinutritional parameters in dry bean genotypes grown in Greece, any studies on the quality of Greek genotypes (varieties and/or landraces) grown in different areas in Greece are added in the text and the references.

  1. Write descriptions for the Figure 5 series.

-Descriptions have been added for these Figures.

  1. Elaborate the conclusions as compared to your article title.

-The Conclusions section has been changed as follows

The effect of genotype, agronomic conditions and environment on the nutritional and anti-nutritive properties of locally adapted Phaseolus vulgaris races was the subject of this study.

From the overall analysis, protein and starch content were mainly affected by agronomical conditions, namely soil type, irrigation and fertilization practices, and marginally affected by genotype and weather conditions, whereas lipid and ash contents on dry beans were less affected by agronomical conditions. Thus, such compounds may be used as a good index for genotype discrimination. Fiber appeared to be affected by both genotype and agronomical conditions, while phenolic and trypsin inhibitors contents were considerably affected by stress conditions either in spotted points or during the whole cultivation period distressing mostly the crop. Due to the organic cultivation methods applied, concentrations of toxic metals were only determined in traces that did not exceed established limits.

Round 2

Reviewer 1 Report

The revision has largely improved the quality of the manuscript.

Author Response

-We would like to thank the reviewer for his/her comments.

Reviewer 2 Report

I still have two questions that have not been solved by the authors after the revision.

1. The authors said that they have done statistical analysis on the data; however, I have not seen any changes on the figures or tables. In addition, the style that the data were expressed in table 7 may can be improved, such as 1.16±0.07.

2. The error bars in Figure 6 still remain the same for the same index. How many replicates did the authors use? The error bars were expressed by standard errors or standard deviation? 

Author Response

  1. The authors said that they have done statistical analysis on the data; however, I have not seen any changes on the figures or tables. In addition, the style that the data were expressed in table 7 may can be improved, such as 1.16±0.07.

The expression of the data in Table 7 has been altered, according to your suggestion.

  1. The error bars in Figure 6 still remain the same for the same index. How many replicates did the authors use? The error bars were expressed by standard errors or standard deviation?

Thank you for your comment. Three replicates were used and the error bars represent standard deviations.

Reviewer 3 Report

the authors have revised it well

Author Response

(The authors gave the same response as above.)
